# A Low-Threshold Miniaturized Plasmonic Nanowire Laser with High-Reflectivity Metal Mirrors

**DOI:** 10.3390/nano10101928

**Published:** 2020-09-27

**Authors:** Jiahui Zheng, Xin Yan, Wei Wei, Chao Wu, Nickolay Sibirev, Xia Zhang, Xiaomin Ren

**Affiliations:** 1State Key Laboratory of Information Photonics and Optical Communications, Beijing University of Posts and Telecommunications, Beijing 100876, China; jiahuizheng@bupt.edu.cn (J.Z.); wuchao920073513@bupt.edu.cn (C.W.); xzhang@bupt.edu.cn (X.Z.); xmren@bupt.edu.cn (X.R.); 2School of Mechanical and Electric Engineering, Guangzhou University, Guangzhou 510006, China; wei@gzhu.edu.cn; 3Photonics Research Centre, Department of Electronic and Information Engineering, The Hong Kong Polytechnic University, Hung Hom, Kowloon 999077, Hong Kong, China; 4ITMO University, Kronverkskiy pr. 49, 197101 St. Petersburg, Russia; nicksibirev@yandex.ru

**Keywords:** plasmonic nanowire laser, reflectivity-enhanced, nanolaser, GaAs

## Abstract

A reflectivity-enhanced hybrid plasmonic GaAs/AlGaAs core-shell nanowire laser is proposed and studied by 3D finite-difference time-domain simulations. The results demonstrate that by introducing thin metal mirrors at both ends, the end facet reflectivity of nanowire is increased by 30–140%, resulting in a much stronger optical feedback. Due to the enhanced interaction between the surface charge oscillation and light, the electric field intensity inside the dielectric gap layer increases, resulting in a much lower threshold gain. For a small diameter in the range of 100–150 nm, the threshold gain is significantly reduced to 60–80% that of nanowire without mirrors. Moreover, as the mode energy is mainly concentrated in the gap between the nanowire and metal substrate, the output power maintains >60% that of nanowire without mirrors in the diameter range of 100–150 nm. The low-threshold miniaturized plasmonic nanowire laser with simple processing technology is promising for low-consumption ultra-compact optoelectronic integrated circuits and on-chip communications.

## 1. Introduction

The minimization of semiconductor laser dimensions is critical to realize ultra-compact photonic integrated circuits. Thanks to the nanoscale dimension, high refractive index, and natural resonant cavity structure, semiconductor nanowires (NWs) have shown great potential in ultra-small lasers. To date, NW lasers from a number of different semiconductors have been demonstrated [1,2,3,4,5,6,7,8,9,10]. However, these photonic NW lasers (PHO laser for short) are restricted in optical mode size and physical device dimension caused by the optical diffraction limit [11,12]. To further reduce the size of NW lasers, plasmonic NW lasers have been proposed [13,14,15]. Surface plasmon resonance (SPR) is defined as an optical phenomenon arising from the collective oscillation of conduction electrons in metal when the electrons are disturbed from their equilibrium positions [16]. There are mainly two types of surface resonances including propagating surface plasmon polariton (SPP) and non-propagating localized SPR (LSPR) [17]. Resonant excitation of LSPR has shown potential in sensing, optical switching, imaging, nonlinear optics, and photodetections [18,19,20]. SPP enables subwavelength optical propagation attributed to its intrinsically two-dimensional restrictions. Placing the NW on a metal substrate to produce SPP modes can localize light, but ohmic losses severely degrade the device performance [21]. A nanometer scale insulating gap between the NW and metal substrate can get lower intrinsic ohmic losses of metals, allowing coupling between the plasmonic and photonic modes across the dielectric gap and effective subwavelength transmission in non-metallic regions with strong mode confinement [22,23,24]. As the plasmonic modes have longer cutoff wavelengths, they are able to downscale the laser beyond the diffraction limit. To date, SPP NW lasers based on various materials and structures have been demonstrated or proposed, including CdS NW [25], GaN NW [26], ZnO NW [27], perovskite NW [28], GaAs NW [29], AlGaAs/GaAs NW/quantum well [30], and GaAs/InGaAs NW/quantum dots [31]. Although these SPP lasers have great potential in ultra-compact photonic integrated circuits, the threshold, dimension, and operating temperature still need to be improved. One limiting factor is the high absorptive loss of the metal. Some effort has been made to reduce the absorptive loss of the metal, such as using high-quality metal film [32], introducing an air gap between the NW and substrate [33], and adopting low-loss higher order SPP mode [34]. The end facet reflectivity also plays an important role in the NW laser performance and dimension [35]. It has been reported that the end facet reflectivity of NWs quickly drops as the diameter decreases in small diameters [10]. As the SPP NW laser typically operates at small diameters, the low reflectivity significantly limits the threshold. However, so far there have been few reports on enhancing the end facet reflectivity of SPP NW lasers.

In this paper, a miniaturized hybrid plasmonic NW laser with high-reflectivity gold mirrors on both ends is proposed and studied. The device is composed of a GaAs/AlGaAs NW placed on a silver substrate separated by a nanoscale MgF_2_ gap layer. Three-dimensional finite-difference time-domain (FDTD) is used to analyze the modal distributions and lasing characteristics of both photonic and plasmonic modes. The results emphasize that the proposed structure can significantly reduce the threshold of the NW laser while maintaining relatively high extraction efficiency, paving the way for the development of low-threshold miniaturized nanolasers and low-consumption ultra-compact optoelectronic integrated circuits. 

## 2. Methods 

Well-facet NWs support predominantly axial Fabry-Perot waveguide modes [36]. The end facet reflectivity determines the NW cavity quality to some degree and further affects the threshold of NW lasers. It has been demonstrated that the gold particle or silver coating at cavity facets can highly increase the reflectivity [10,37]. However, for photonic lasers, the metal particle or coating would dramatically degrade the laser extraction efficiency [10]. For SPP lasers, as the mode field energy is mainly concentrated in a thin insulating gap between the NW and the metal substrate, the threshold is expected to be reduced by increasing the end facet reflectivity without significantly degrading the extraction efficiency level. In this study, four types of NW lasers were simulated: the PHO laser, PHO laser with gold mirrors (PHOG), SPP laser, and SPP laser with gold mirrors (SPPG). For PHO and PHOG lasers, the NWs were placed on a SiO_2_/Si substrate. For SPP and SPPG lasers, the NWs were placed on a MgF_2_/Ag substrate. MgF_2_ is one of the lowest index materials in infrared with a useful transmission range from 0.19 um to 6.5 um [38]. When reducing the NW diameter, the mode can be strongly confined in two dimensions within the MgF_2_ gap, providing a larger propagation length than that of the equivalent metal-semiconductor interface [22]. Two 30 nm thickness gold layers were placed on both ends of the NWs, which act as high-reflectivity mirrors. 

The schematic diagram of the SPPG NW laser is shown in Figure 1. The NW was composed of GaAs/AlGaAs core-shell heterostructure with a length of 6 μm. The 10 nm AlGaAs shell with an Al composition of 32% acted as a passivation layer to improve the radiative efficiency [10]. The thickness of the MgF_2_ layer was 5 nm. The NW has a hexagonal cross section and only one side facet was contacted with the substrate. The complex refractive indices for GaAs were taken from Palik [39]. The optical constants of the gold mirrors and silver substrate and the refractive index of MgF_2_ were obtained from [40] and [38], respectively. Electromagnetic field distributions and lasing characteristics of NW lasers were calculated by Lumerical FDTD solutions with perfectly matched layers as boundary conditions. Mode source with a center wavelength of 875 nm was used, corresponding to the band-edge emission of GaAs at room temperature. We utilized 3D frequency-domain profile and power monitors as analysis groups to calculate the lasing performance parameters. The mesh step was set to as small as 1 nm in vertical direction due to the ultra-small thickness of the MgF_2_ layer.

## 3. Results and Discussion

Figure 2a presents the cross-section electric field distribution of fundamental supported guided modes in the PHO laser with a diameter of 300 nm, which are identified as HE11_y_, HE11_x_, TE01, and TM01 modes, respectively. Figure 2b compares the propagating HE11_y_ mode profile of PHO and PHOG lasers near the NW output end. It should be noted that all of the electric field intensity is normalized by the maximum electric field intensity obtained in the structures. In this way, the electric field intensity enhancement by the metal mirrors can be directly watched. It can be seen that the guiding modes were strongly confined inside the NW due to the large dielectric contrast between NW and air. As the confinement of modes is dominated by the refractive index between NW and side surroundings, the gold mirrors have little influence on the lateral mode distribution. However, the enhanced end facet reflectivity significantly increases the optical feedback, resulting in much stronger electric field intensity, as shown in Figure 2b. Figure 3a describes the diameter-dependent reflectivity of modes in PHO and PHOG lasers. Due to the low refractive index of gold in the near-infrared region, the refractive index difference between the GaAs NW and gold mirror is much larger than that between the NW and air, resulting in significantly higher reflectivity for most modes. For example, at a diameter of 300 nm, the reflectivity of the HE11_y_ mode exceeds 0.7 in the PHOG laser, almost twice that in the PHO laser. The only exception is the TM01 mode, which exhibits lower reflectivity for the PHOG laser at large diameters. Figure 3c,d give the mode reflection at the NW end facets for TE01 and TM01 mode in a 300 nm diameter GaAs NW, respectively. For the TE01 mode, the reflected optical power is increased at the NW/gold interface, which is because the TE01 mode has electric field components parallel to the NW/gold interface and doesn’t couple to the plasmonic modes of the gold mirror. For the TM01 mode, the reflected optical power is much lower, which is attributed to the electric field components perpendicular to the NW/gold interface, resulting in a strong coupling of TM01 mode to the plasmonic modes of the gold mirror. Hence, for NW lasers lasing with TM01 mode, the gold mirrors would significantly degrade the overall performance [10]. Moreover, although gold mirrors could improve the reflectivity of most modes, as most of the light power in the PHO lasers is extracted from the NW end facet, the metal mirrors would significantly degrade the extraction efficiency. Restricted by the optical diffraction limit, the diameter of PHO lasers is relatively large. For example, to achieve low-threshold lasing, GaAs NW should have a diameter of at least 330 nm [10]. From Figure 3b we can see that the extraction efficiency of PHOG lasers rapidly decreases as the diameter increases. For example, at a diameter of 300 nm, the output power of HE11_y_ mode decreases by nearly 80% after introducing the metal mirrors. Hence, although the introduction of metal mirrors is expected to reduce the threshold by increasing the end facet reflectivity of most modes, the large drop of extraction efficiency makes it impractical in PHO lasers.

Now we turn to investigate the influence of gold mirrors on the performance of plasmonic NW lasers. SPP mode is generated by the resonant interaction between the surface charge oscillation and the y-polarized electric field of the light [41]. The electromagnetic energy of SPP mode is distributed over both the NW and the adjacent metal-dielectric interface at large diameters, and strongly confined in two dimensions within the dielectric gap layer when reducing the diameter. Figure 4a shows the electric field distributions of modes in SPP NW lasers. It can be seen that most of the power of HE11_y_ mode is strongly confined in the MgF_2_ layer between the NW and metal film, corresponding to a long-range plasmonic polarization subwavelength transmission with low loss. The HE11_x_ mode is confined inside the NW as it is x-polarized and cannot excite SPP modes. As the TE01 and TM01 modes have both x-polarization and y-polarization, they exhibit a mixture characteristic of photonic mode and plasmonic mode. Similar to the photonic NW lasers mentioned above, the gold mirrors have little influence on the lateral mode distribution but strongly affect the propagating mode profile. Figure 4b presents the profile of the propagating HE11_y_ mode intensity of SPP and SPPG lasers near the NW output end, both of which are normalized by the maximum electric field intensity in the SPPG laser. It can be obviously seen that the electric field intensity inside the MgF_2_ layer is significantly enhanced after introducing gold mirrors. Due to the increased end facet reflectivity, the optical feedback inside the NW is enhanced. This means that the interaction between the surface charge oscillation and light is enhanced, resulting in stronger electric field intensity inside the MgF_2_ layer. 

To investigate the guiding and lasing properties of the SPP lasers, dependences of the end facet reflectivity, modal confinement (CF), modal loss, and quality factor (Q factor) of the NW diameter are calculated and presented in Figure 5. Figure 5a shows the diameter-dependent end facet reflectivity of SPP and SPPG lasers. The reflectivity of both SPP and SPPG lasers first increases and then decreases as the NW diameter increases. After introducing gold mirrors, the reflectivity of all modes was significantly improved. For example, at a small diameter of 100 nm, the reflectivity of the HE11_y_ mode increased from less than 0.3 to close to 0.5. In comparison with the PHO lasers, the reflectivity enhancement in SPP lasers was slightly lower. This is because only a portion of the SPP mode energy overlapped with the gain materials. When the diameter increased to 160 nm, the reflectivity of SPPG laser rapidly increased, similar to the trend of PHOG laser, because the proportion of energy in NW increased. Overall, the gold mirrors significantly enhanced the end facet reflectivity and offer a new way to optimize the performance of SPP lasers. 

The CF represents the coupling efficiency between the gain medium and resonate modes. CF for a dielectric waveguide is given by [10]:(1)Γ=cε0na∬s12|E|2dxdy∬∞Re[(E×H)z⋅dxdy]
where E and H are the complex electric and magnetic fields of the waveguide modes, respectively, c is the vacuum speed of light, $\varepsilon_{0}$ is the vacuum permittivity, and $n_{a}$ is the refractive index of the gain medium at a particular frequency ω of the guided mode. As shown in Figure 5b, the CF of SPP and SPPG laser is positively correlated with the NW diameter and is approximately the same, except for small fluctuations caused by the competition between SPP and PHO modes.

The modal loss per unit length $\alpha_{m}$ is related to the bulk material loss $\left(\alpha_{0}^{m}=\omega \varepsilon {\prime \prime }_{m}/cn_{m}\right)$ and can be obtained by [42]:(2)$$\alpha_{m}^{}=\frac{n_{g}}{n_{m}}\frac{\iint_{m}{\left|E\right|}^{2}dxdy}{\iint_{\infty }{\left|E\right|}^{2}dxdy}a_{0}^{m}$$
where $n_{m}$ is the refractive index of the metal substrate at a particular frequency ω of the guided mode and $n_{g}=c/v_{g}$, where *c* is the speed of light in vacuum. As presented in Figure 5c, the modal loss of HE11_y_ and TE01 modes first increases and then decreases with the increase of diameter, while the loss of HE11_x_ reveals the opposite trend. This is an interesting description of the coupling process between the y-polarized wave and the metal. For HE11_y_ mode, the energy is strictly confined in the dielectric layer in the diameter range from cut-off to 120 nm. When the diameter exceeds 120 nm, the mode energy center gradually moves upwards with the increase of diameter, leading to a decrease of energy trapped in the metal substrate and reduction of modal loss. For the x-polarized wave like HE11_x_ mode, the lateral confinement increases with the increase of diameter until 220 nm, after which more and more energy leaks from NW to the substrate.

The *Q* factor of a mode that is indicative of how long the stored energy of that mode remains in the NW cavity is defined as [43]:(3)$$Q=\frac{\omega }{\bar{v_{g}}\left[\alpha_{m}^{}+\frac{1}{L}\ln\left(\frac{1}{R_{}}\right)\right]}$$
which accepts the influence of modal absorption loss and mirror loss. As illustrated in Figure 5d, *Q* factor is positively correlated with the NW diameter. All modes exhibit a higher *Q* factor in SPPG laser than SPP laser due to the enhanced reflectivity. A higher *Q* factor typically means that the cavity mode has more chance to lase, which plays an important role in judging the feasibility of SPPG lasers.

Lasing threshold is the lowest excitation level at which laser output is dominated by stimulated emission rather than spontaneous emission. The threshold gain which describes the required gain per unit length for lasing is defined as [42]:(4)$$g_{th}=\frac{1}{\Gamma }\left(\alpha_{m}+\frac{1}{2L}\ln\frac{1}{R_{1}R_{2}}\right)$$
where L is the length of NW and $R_{1}$ and $R_{2}$ is the reflectivity at the NW end facet, assuming $R_{1}=R_{2}$. The dependence of threshold gain on diameter is presented in Figure 6a. Obviously, HE11_y_ mode in the SPPG laser exhibited much lower threshold gain in comparison with that in the SPP laser. At a diameter of 100 nm near the cut-off, the threshold gain decreased from 7060 to 4020 cm^−1^ after introducing gold mirrors, demonstrating that lasing with an ultra-small size is much easier for the SPPG laser. The ultra-small threshold gain makes the SPPG laser particularly promising for minimized optoelectronic integrated circuits. 

The output power is also an important factor of lasers. As mentioned earlier, the metal mirrors can significantly enhance the NW end reflectivity but dramatically reduce the extraction efficiency of PHO lasers, degrading the overall performance of PHO lasers. To study the influence of metal mirrors on the output power of SPPG laser, the output powers of SPP and SPPG lasers are compared, as shown in Figure 6b. At a small diameter range of 100–150 nm, the output HE11_y_ mode power of the SPPG laser maintains > 60% that of the SPP laser, as most energy of HE11_y_ mode is concentrated in the MgF_2_ gap layer and the gold mirrors have little influence on the output power. When the diameter increases, the extraction efficiency rapidly drops due to the increase of modal energy overlap with NW. Hence for a miniaturized plasmonic NW laser, the gold mirrors can significantly improve the threshold gain while maintaining a relatively high output power. 

## 4. Conclusions

In summary, a plasmonic GaAs/AlGaAs core-shell NW laser with high-reflectivity gold mirrors was proposed and studied. After introducing gold mirrors, the end facet reflectivity was substantially enhanced, resulting in much stronger optical feedback inside the NW. Due to the enhanced interaction between the surface charge oscillation and light, the electric field intensity inside the dielectric gap layer increases, resulting in a much lower threshold gain. As the mode energy is mainly concentrated in the gap between the NW and metal substrate, the extraction efficiency remains high, particularly at small diameters. In this work, Au is adopted as a mirror due to its low absorption coefficient at near-infrared wavelength range [40,44]. However, as the metal mirrors have little influence on the output power of SPP lasers, other cheaper metals such as Al and Ag are also applicable. Moreover, other reflection structures, such as distributed Bragg reflectors, may also be suitable as mirrors due to their high reflectivity and low absorption loss [45]. This work provides a simple way to reduce the threshold gain of SPP NW lasers at an ultra-small diameter. 

## Figures and Tables

**Figure 1 nanomaterials-10-01928-f001:**
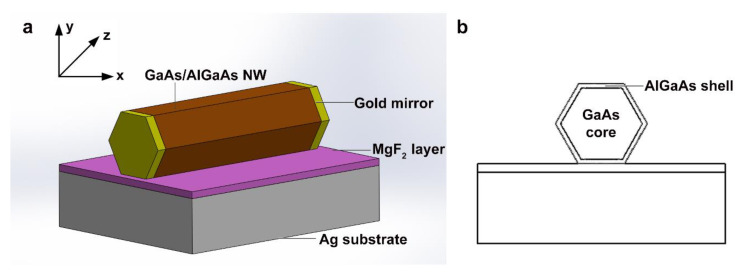
Schematic diagrams of the surface plasmon polariton laser with gold mirrors (SPPG). (**a**) 3D model and (**b**) sectional view.

**Figure 2 nanomaterials-10-01928-f002:**
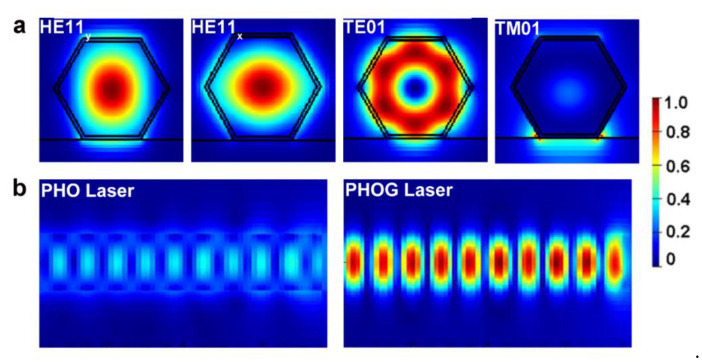
(**a**) The lateral electric field distribution of modes in PHO lasers. (**b**) The profile of the propagating HE11_y_ mode of PHO and PHOG lasers near the NW output end. The diameter is 300 nm.

**Figure 3 nanomaterials-10-01928-f003:**
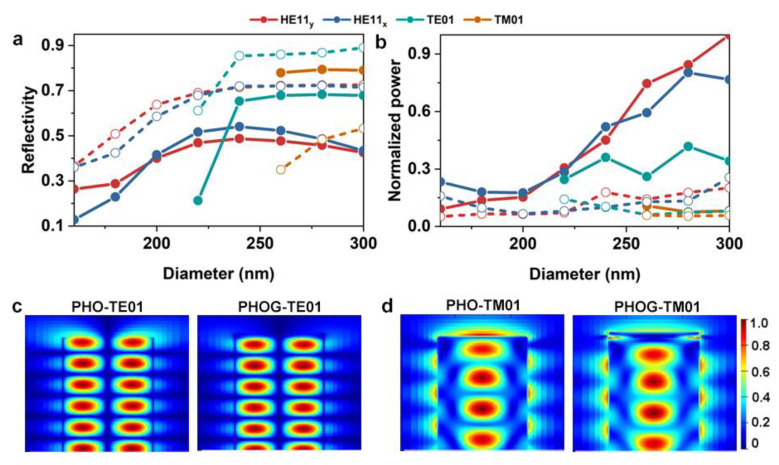
(**a**) Diameter-dependent end facet reflectivity and (**b**) normalized output power of PHO (solid line) and PHOG (dash line) lasers. Electric field intensity profiles at the NW/air interface and NW/Au interface for TE01 (**c**) and TM01 (**d**) mode respectively. The diameter is 300 nm.

**Figure 4 nanomaterials-10-01928-f004:**
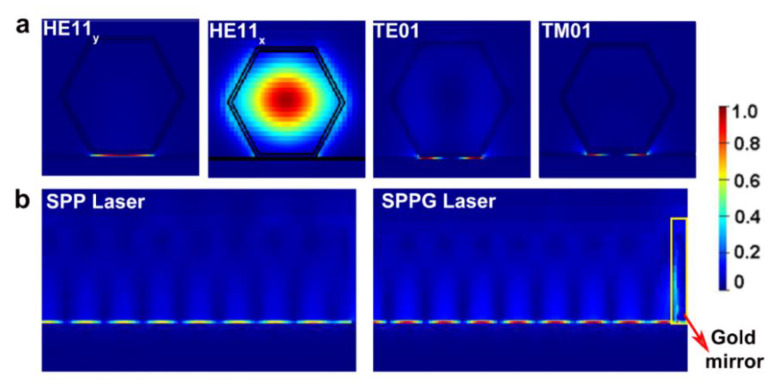
(**a**) The lateral electric field distribution of modes in SPP laser. (**b**) The profile of the propagating HE11_y_ mode of SPP and SPPG laser near the NW output end. The diameter is 100 nm.

**Figure 5 nanomaterials-10-01928-f005:**
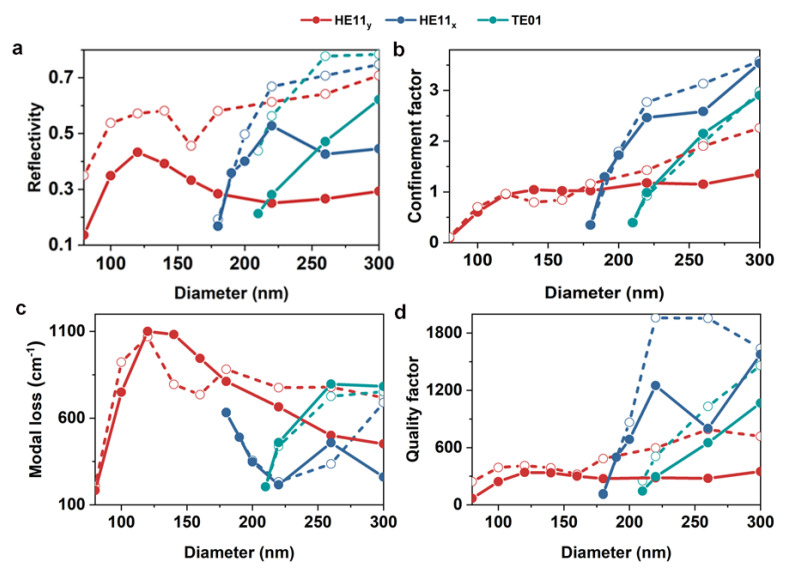
End facet reflectivity (**a**) and modal confinement (CF) (**b**) versus NW diameter for the SPP Laser (solid lines) and SPPG Laser (dash lines). (**c**,**d**) Modal loss and Q factor versus NW diameter for the SPP Laser (solid lines) and SPPG Laser (dash lines).

**Figure 6 nanomaterials-10-01928-f006:**
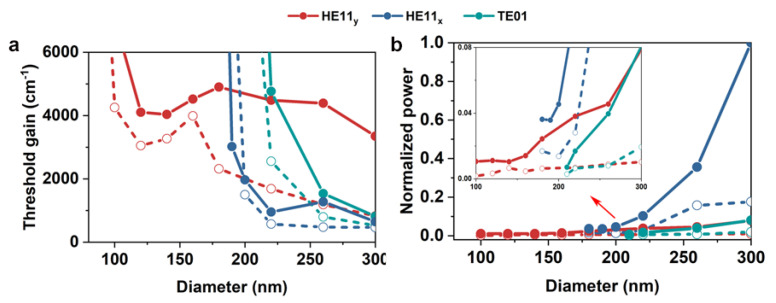
(**a**,**b**) Threshold gain and normalized output power for SPP (solid lines) and SPPG (dash lines) lasers. The inset in (**b**) infers the normalized output power of HE11_y_ mode.

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
