# Peer review of "A Low-Threshold Miniaturized Plasmonic Nanowire Laser with High-Reflectivity Metal Mirrors"

_nanomaterials, 2020, doi:10.3390/nano10101928_

Round 1
Reviewer 1 Report
The authors report on the influence of high-reflectivity metal mirrors on the performance of hybrid plasmonic GaAs/AlGaAs core-shell nanowire lasers based on their simulation results. The investigation is interesting in view of ongoing miniaturization efforts for optical devices. I recommend the manuscript for publication provided that the following comments are addressed:
1) The authors should comment on the possibility of using other metals than Au, considering that cost effectiveness is important for the use in integrated circuits and Au cannot be introduced in all processing lines.
2) Information on simulation should be more detailed.
Reviewer 2 Report
In this work, "A low-threshold miniaturized plasmonic nanowire laser with high-reflectivity metal mirrors", the authors numerically studied a reflectivity-enhanced hybrid plasmonic GaAs/AlGaAs core-shell nanowire laser. Based on the obtained results, the authors claimed that the proposed technique is promising for low-consumption ultra-compact optoelectronic integrated circuits and on-chip communications. Overall, this manuscript has a strong potential for a second review after applying the issues and addressing the shortcomings listed below:
1-The authors should polish/revise some grammatical mistakes and typos along the manuscript. For example, “show” is used too many times. Try to consider using synonyms. From now on, I am not mentioning the remaining grammatical mistakes/typos along the manuscript. I invite the authors to read their manuscript carefully and make the required changes where necessary.
2-Revise the following statement: ‘However, there have been little reports on…’. From now on, I am not mentioning the remaining grammatical mistakes/typos along the manuscript. I invite the authors to read their manuscript carefully and make the required changes where necessary.
3-In Figure 1a, make the length of the arrow for z-axis closer to the others. Besides, make it orthogonal to other axes and mention the names of the remaining materials as well.
4-In the Introduction section, while discussing plasmonics and its possible applications, the following works should also be considered and cited, to give a more general view to the possible readers of the work: [(i) Monolithic metal dimer-on-film structure: new plasmonic properties introduced by the underlying metal, Nano Letters 20, 2087-2093 (2020); (ii) The role of Ge2Sb2Te5 in enhancing the performance of functional plasmonic devices, Materials Today Physics 12, 100178 (2020); (iii) Functional charge transfer plasmon metadevices, Research 2020, 9468692 (2020)].
5-For the utilized numerical tool (Lumerical FDTD), more detailed information should be provided for the possible readers of the study.
6-In Figure 2 and Figures 3a,b, the provided maps are for “E” or “E/E0” (E-field enhancement)? Please explain and try to plot E-field enhancements (in addition to the current plots).
7-More information should be given for the use of MgF2, rather than just saying “…supports low-loss propagation under…”. Likewise, where did you get the optical constants for GaAs/AlGaAs and MgF2? The corresponding references should be mentioned within the manuscript.
Round 2
Reviewer 2 Report
In its current form, the revised manuscript is suitable for publication.